# Application of Kampo Medicines for Treatment of General Fatigue Due to Long COVID

**DOI:** 10.3390/medicina58060730

**Published:** 2022-05-28

**Authors:** Kazuki Tokumasu, Keigo Ueda, Hiroyuki Honda, Naruhiko Sunada, Yasue Sakurada, Yui Matsuda, Yasuhiro Nakano, Toru Hasegawa, Yuki Otsuka, Mikako Obika, Hideharu Hagiya, Hitomi Kataoka, Fumio Otsuka

**Affiliations:** 1Department of General Medicine, Okayama University Graduate School of Medicine, Dentistry and Pharmaceutical Sciences, Okayama 700-8558, Japan; tokumasu@okayama-u.ac.jp (K.T.); philopotchy@live.jp (K.U.); hydrogen77@me.com (H.H.); naru.kun.red.1117@gmail.com (N.S.); sakurada202@gmail.com (Y.S.); m05089ym@gmail.com (Y.M.); y-nakano@okayama-u.ac.jp (Y.N.); t.hase5178.1@gmail.com (T.H.); otsuka@s.okayama-u.ac.jp (Y.O.); obika-m@cc.okayama-u.ac.jp (M.O.); hagiya@okayama-u.ac.jp (H.H.); hitomik@md.okayama-u.ac.jp (H.K.); 2Clinical and Educational Center for Kampo Medicine, Okayama University Hospital, Okayama 700-8558, Japan

**Keywords:** general fatigue, herbal medicine, Kampo medicine, long COVID

## Abstract

Evidence regarding treatment for the acute phase of COVID-19 has been accumulating, but specific treatment for long COVID/post-COVID-19 condition has not yet been established. Treatment with herbal medicine might be one treatment option for long COVID, but there has been little research on the effectiveness of herbal medicine for long COVID. The aim of this study was to clarify the prescription patterns of Kampo medicines, which are herbal medicines that originated in China and were developed in Japan, for the treatment of general fatigue due to long COVID. A retrospective descriptive study was performed for patients who visited a COVID-19 aftercare clinic established in Okayama University Hospital during the period from Feb 2021 to Dec 2021 with a focus on symptoms accompanying general fatigue and prescriptions of Kampo medicine. Among the clinical data obtained from medical records of 195 patients, clinical data for 102 patients with general fatigue and accompanying symptoms were analyzed. The patients had various symptoms, and the most frequent symptoms accompanying general fatigue were dysosmia, dysgeusia, headache, insomnia, dyspnea, and hair loss. Prescriptions of Kampo medicine accounted for 24.1% of the total prescriptions (*n* = 609). The most frequently prescribed Kampo medicine was hochuekkito (71.6%) and other prescribed Kampo medicines were tokishakuyakusan, ryokeijutsukanto, juzentaihoto, hangekobokuto, kakkonto, ninjin’yoeito, goreisan, rikkunshito, and keishibukuryogan. Since the pathophysiology of general fatigue after an infectious disease is, in general, considered a qi deficiency in Kampo medicine, treatments with such compensation agents can be the major prescription as a complement for the qi. In conclusion, Kampo medicine can be one of the main pharmacological treatments for long COVID accompanying general fatigue.

## 1. Introduction

The pandemic of novel coronavirus disease 2019 (COVID-19) has been continuing for more than 2 years, and more than 285 million cases and 5.4 million deaths have been reported worldwide [1]. In addition to the acute-phase symptoms induced by a viral infection, COVID-19 can also cause prolonged sequelae, which have recently been termed long COVID or post-acute sequelae of SARS-CoV-2 infection (PASC) and have recently been defined as the post-COVID-19 condition by the World Health Organization [2,3,4]. Various symptoms of long COVID including general malaise, dysosmia, dysgeusia, low-grade fever, headache, and hair loss have been reported worldwide [4,5,6] and in Japan [7,8,9]. It has been reported that approximately one-third of COVID-19 patients suffer from those sequelae symptoms, and the sequelae symptoms are still important clinical problems [10,11]. However, the pathophysiology of long COVID remains unclear and an effective treatment strategy for long COVID has not yet been established.

General fatigue has been reported to be the most common symptom of long COVID [6,8] and general fatigue in some patients progresses to myalgic encephalomyelitis/chronic fatigue syndrome (ME/CFS) [12,13,14]. ME/CFS is a debilitating illness with a wide range of symptoms including post-exertional malaise, unrefreshing sleep, and cognitive impairment [15,16]. ME/CFS is occasionally a severe and potentially long-lasting condition and it is a socially and medically important disease [17]. Previous studies have shown that pharmacological therapies such as glucocorticoids, cytokine inhibitors, antibiotics, and behavioral therapies such as cognitive behavioral therapy and graded exercise therapy could be effective in treating ME/CFS in some limited settings [18,19]. Because of the unclear etiology, diagnostic uncertainty, and the resultant heterogeneity of the chronic fatigue syndrome population, there are no firmly established treatment recommendations for ME/CFS.

Given the limited treatment options offered by conventional Western medicine, there have been reports on the use of complementary and alternative medicines for ME/CFS [16,20] as well as Kampo medicine (Japanese traditional medicine), which originated in China and was developed in Japan. Despite the potential use of Kampo medicine for acute and chronic phases of COVID-19 [21,22], there has been little research focusing on Kampo medicine. Kampo medicine can provide new perspectives on treatment for post-infectious fatigue and for general fatigue in patients with long COVID. The aim of this study was to characterize general fatigue symptoms related to long COVID and to clarify the prescription patterns of Kampo medicine.

## 2. Patients and Methods

### 2.1. Study Design

This study was a descriptive study conducted in a single facility. A COVID-19 aftercare (CAC) clinic was established on 15 Feb 2021 in the Department of General Medicine of Okayama University Hospital, a tertiary hospital with 865 beds located in the western area of Japan. The purpose of establishing the CAC clinic was to evaluate and manage patients who suffer from symptoms of long COVID. The clinic handles patients who had a sequela even for only one month (four weeks) after the onset of COVID-19 and patients referred from other medical facilities.

### 2.2. Patients’ Characteristics

Clinical information was obtained retrospectively for patients who visited our CAC clinic. Medical records for 195 patients with long COVID who visited the CAC clinic during the period from Feb 2021 until Dec 2021 were carefully reviewed. Long COVID was defined as symptoms that persist for more than one month after the onset of COVID-19 [2,3,4]. Information was obtained from medical records for age, gender, body mass index (BMI), severity of the acute phase of COVID-19 [23], duration from the onset of COVID-19 to the first visit to the CAC clinic, history of COVID-19 vaccination, and clinical symptoms of long COVID. We included patients who reported fatigue as a subjective symptom in our study. Data for 102 of the 195 patients were used for analysis.

### 2.3. Background of Physicians Working in the CAC Clinic

The physicians who prescribed the Kampo medicine were approximately 10 Japanese well-trained general practitioners who had graduated from a Japanese medical school and had completed two years of general medicine training. The physicians had been practicing general medicine, including Kampo medicine, for 5 to 30 years after graduation. Any physician can consult with doctors specialized in Kampo medicine and all these doctors often discussed with each other about the patients at the long COVID conferences.

### 2.4. Trends of Kampo Prescription

We included medications that were being taken by patients at the time of referral to the CAC clinic and prescriptions during management at the clinic. We excluded eight prescriptions due to duplication.

### 2.5. Ethical Approval

Information regarding the present study was provided on our hospital wall and on the website of our hospital, and patients who wished to opt out were offered that opportunity. Informed consent from the patients was not necessary due to the anonymization of data. This study was approved by the Ethics Committee of Okayama University Hospital (No. 2105-030) and adhered to the Declaration of Helsinki.

## 3. Results

Data for all of the 195 patients visiting our CAC clinic during the study period were obtained from medical records. We focused on symptoms accompanying general fatigue and prescriptions of Kampo medicine. Data for 102 patients were used for analysis. The clinical backgrounds of the patients visiting our CAC clinic are shown in Table 1. The 102 patients included 47 males (46.1%) and 55 females (53.9%). The median BMI of the patients was 22.9 (IQR: 20.8–26.6). The number and proportion of patients for each duration after the onset of COVID-19 to the first visit to the CAC clinic were 32 and 31.4% for 1–2 months, 26 and 25.5% for 2–3 months, 19 and 18.6% for 3–4 months, 11 and 10.8% for 4–5 months, 3 and 2.9% for 5–6 months, and 11 and 10.8% for more than 6 months.

As for the severity of COVID-19 defined by the Ministry of Health, Labour and Welfare in Japan [23], the numbers (proportions) of patients with mild, moderate-I, moderate-II, and severe states were 75 (73.5%), 9 (8.8%), 18 (17.6%), and 0, respectively. The numbers (proportions) of patients who did not receive a COVID-19 vaccine, who received one dose of a vaccine, and who received two doses of a vaccine were 66 (64.7%), 13 (12.7%), and 22 (21.6%), respectively. All of the participants were Japanese patients.

The frequencies of symptoms accompanying general fatigue or not at the initial visit are shown in Figure 1. There were more than 10 symptoms of long COVID accompanying general fatigue, and the most frequent symptoms were dysosmia (29 patients, 28.4%), dysgeusia (29 patients, 28.4%), headache (26 patients, 25.5%), insomnia (23 patients, 22.5%), hair loss (20 patients, 19.6%) and dyspnea (19 patients, 18.6%). The most frequent symptoms without general fatigue were dysosmia (43 patients, 46.2%), dysgeusia (36 patients, 38.7%), hair loss (29 patients, 31.2%), headache (12 patients, 12.9%), dyspnea (12 patients, 12.9%), insomnia (5 patients, 5.4%), cough (5 patients, 5.4%) and numbness (5 patients, 5.4%).

The total number of prescriptions was 617 and 8 prescriptions were excluded due to duplication. Of the 609 prescriptions, 147 (24.1%) were Kampo medicine (Table 2). The number of patients receiving each Kampo medicine is shown in Figure 2. The most frequent prescriptions of Kampo medicine were hochuekkito (73 patients, 71.6%), tokishakuyakusan (10 patients, 9.8%), ryokeijutsukanto (10 patients, 9.8%), juzentaihoto (9 patients, 8.8%), hangekobokuto (7 patients, 6.8%), ninjin’yoeito (5 patients, 4.9%), kakkonto (4 patients, 3.9%), rikkunshito (4 patients, 3.9%), goreisan (3 patients, 2.9%) and keishibukuryogan (3 patients, 2.9%).

The frequencies of Kampo medicines prescribed for the six most common symptoms (dysosmia, dysgeusia, insomnia, headache, dyspnea, and hair loss) associated with fatigue are shown in Figure 3. Hochuekkito, tokishakuyakusan, ryokeijutsukanto, juzentaihoto, hangekobokuto, kakkonto, ninjin’yoeito, goreisan, rikkunshito and keishibukuryogan were prescribed for various symptoms.

## 4. Discussion

This study revealed the trends of prescriptions of Kampo medicine for patients who visited our CAC clinic that is specialized in managing the long COVID/post-COVID-19 condition. The characteristics of the patients are shown in Table 1. Most of the patients were in their teens to fifties. There were slightly more female patients than male patients. The median BMI was lower than that reported abroad [24]. The severity of COVID-19 in the acute phase was mild in three-quarters of the patients, and there were no severely ill patients. The number of patients decreased with prolongation of the duration after the onset of COVID-19 to the first visit to the CAC clinic, though about 11% of the patients were seen more than six months after the onset of the disease. The most frequent symptoms accompanying general fatigue were dysosmia and dysgeusia (Figure 1). It has been reported that accompanying symptoms including smelling and tasting disorders tended to be prolonged after COVID-19 [7]. Dysosmia, dysgeusia, and hair loss were prominent symptoms in the long COVID patients without fatigue (Figure 1) since these symptoms tend to remain for a long time after general fatigue improves [7,8].

The present study focused on the prescription trends of Kampo medicines in patients with the long COVID/post-COVID-19 condition who had general fatigue. This is because Kampo medicines are often used for non-specific and comprehensive conditions such as generalized malaise [25]. In contrast to conventional Western medicine, the approach in Kampo medicine to health and healing includes the concept that diseases are the result of imbalance and disharmony between organ systems in the body and between the body and the environment. In addition, blockage or deficiency of the body’s vital energy (known as qi) may result in the occurrence of “illness” [26]. Post-infectious fatigue status has frequently been revealed to be the status of “qi deficiency” [21,22] and hochuekkito is one of the representative “qi-tonifying formulas” among Kampo medicines.

Hochuekkito was the most frequently prescribed Kampo medicine in this study as shown in Figure 2. Hochuekkito is an agent with complex effects including both therapeutic and prophylactic effects and it improves the harmful effect of mental stress on the body. In Japan, the industrially produced Kampo formulation of hochuekkito extract preparation is commonly used for research, and its quality is constant and well-standardized [27]. In 1967, its use was permitted by the Japanese national health insurance system for the treatment of various conditions and diseases including reduced digestive function, anorexia, gastroptosis, and visceral prolapse. It has been reported that hochuekkito is effective for improving conditions of exhaustion and/or frailty. Hochuekkito was also shown to promote locomotor activity in a model of CFS induced by injection of Brucella abortus antigen [28,29]. It has also been reported that hochuekkito showed the effectiveness in animal models of depression through the effects of serotonin [22].

Hochuekkito, kakkonto, and keishibukuryogan were prescribed at similar frequencies for the six symptoms shown in Figure 3. Dysosmia and dysgeusia were more frequent in patients prescribed tokishakuyakusan. It was reported that tokishakuyakusan can prevent the reduction of the dopamine metabolites 3,4-dihydroxyphenylacetic acid and homovanillic acid and increase olfactory bulb nerve growth factor (NGF) in mice that were made anosmic by intranasal infusion of zinc sulfate [30]. Kampo medicines have traditionally been used for non-specific and comprehensive conditions in contrast to a Western medical perspective [31].

In this study, the prescription patterns of Kampo medicine were analyzed only for long COVID patients who visited our CAC clinic. There were several limitations in this study. First, this study was performed in a single center in Japan. Second, the inclusion criteria of the study participants depended on subjective complaints of general fatigue. We did not include patients for whom general fatigue was assessed by a quantitative evaluation tool such as a numerical rating scale or visual analog scale. Third, not all of the physicians who prescribed Kampo medicine necessarily performed all of the assessments for Kampo diagnosis, including tongue and pulse diagnosis, since these are complex conditions, and the diagnosis and prescriptions depend on the physician’s experience. Further studies are needed to predict what proportion of long COVID patients will develop ME/CFS and to determine the relationship between Kampo diagnosis and Kampo prescriptions and how Kampo medicine affects long COVID/post-COVID-19 conditions including ME/CFS.

In conclusion, patients with general fatigue after COVID-19 have various accompanying symptoms, and Kampo medicines are frequently described. Hochuekkito was the most frequently prescribed Kampo medicine for patients with the long COVID/post-COVID-19 condition who visited our CAC clinic.

## Figures and Tables

**Figure 1 medicina-58-00730-f001:**
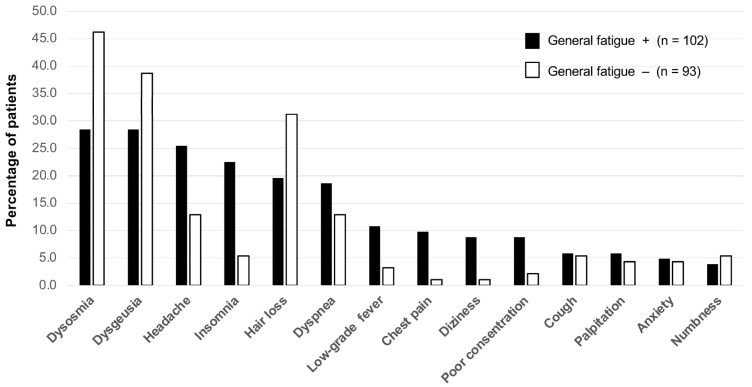
Frequencies of symptoms accompanying and not accompanying general fatigue at the initial visit. The five most frequent symptoms with general fatigue were dyosmia, dysgeusia, headache, insomnia, and hair loss. The five most frequent symptoms without general fatigue were dysosmia, dysgeusia, hair loss, headache, and dyspnea.

**Figure 2 medicina-58-00730-f002:**
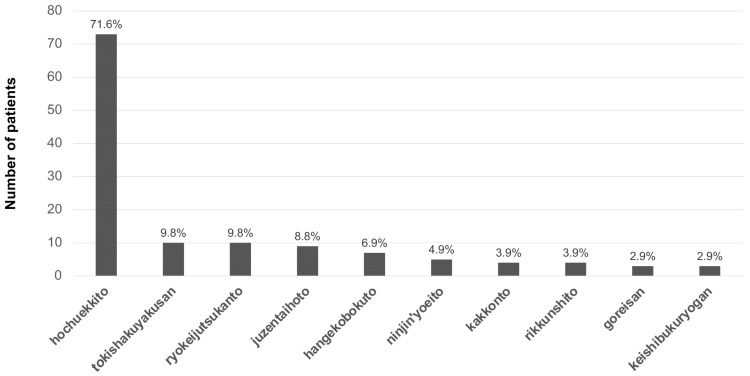
Number of patients receiving each Kampo medicine. The most frequently prescribed Kampo medicine was hochuekkito.

**Figure 3 medicina-58-00730-f003:**
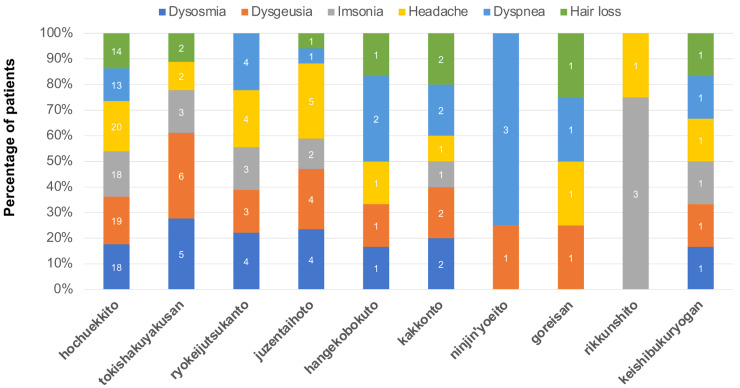
Frequencies of Kampo medicines prescribed for symptoms accompanying general fatigue.

**Table 1 medicina-58-00730-t001:** Backgrounds of 102 patients who visited the COVID-19 aftercare outpatient clinic.

Age Distribution	
**Median (IQR)**	38.5 (24.3–48.8)
<19	13 (12.7%)
20–29	22 (21.6%)
30–39	22 (21.6%)
40–49	21 (20.6%)
50–59	19 (18.6%)
60–69	2 (2.0%)
>70	3 (2.9%)
**Gender**	
Male	47 (46.1%)
Female	55 (53.9)
**BMI**	
Median (IQR)	22.9 (20.8–26.6)
<25	65 (63.7%)
25–30	30 (29.4%)
>30	7 (6.9%)
**Severity of the acute phase of COVID-19**	
Mild	75 (73.5%)
Moderate-Ⅰ	9 (8.8%)
Moderate-Ⅱ	18 (17.6%)
Severe	0
**COVID-19 vaccination**	
None	66 (64.7%)
1 dose	13 (12.7%)
2 doses	22 (21.6%)
unknown	1 (1%)
**Duration after the onset of COVID-19 to the first visit**	
1–2 months	32 (31.4%)
2–3 months	26 (25.5%)
3–4 months	19 (18.6%)
4–5 months	11 (10.8%)
5–6 months	3 (2.9%)
>6 months	11 (10.8%)
**Race**	
Japanese	102 (100%)
Total	102 (100%)

**Table 2 medicina-58-00730-t002:** Number of prescriptions (%) for patients who visited the CAC clinic classified as Kampo medicine and Western medicine with focus on general fatigue.

Types of Medicine	Number of Prescriptions (%)
Kampo Medicine	147 (24.1%)
Western Medicine	462 (75.9%)
Total	609 (100%)

## Data Availability

Information regarding the present study was provided on our hospital wall and on the website of our hospital, and patients who wished to opt out were offered that opportunity.

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
