# Peer review of "Application of Kampo Medicines for Treatment of General Fatigue Due to Long COVID"

_medicina, 2022, doi:10.3390/medicina58060730_

Round 1
Reviewer 1 Report
Maybe there is a description and even translation for the different Kampo-substances used. This would improve interest for the westerns medicine.
Author Response
Major revisions’ points:
Reviewer 1:
Maybe there is a description and even translation for the different Kampo-substances used. This would improve interest for the westerns medicine.
Response: Thank you for your time taken to review our manuscript. We added descriptions of different Kampo substances as supplemental data as follows:
Usage of Kampo substance is based on reference to STORK (Standards of Reporting Kampo Products).
Hochuekkito
This medicine is used for treating the following symptoms in patients who are dispirited, have poor gastrointestinal function and tend to get tired: infirm constitution, fatigue and malaise, declined constitution after disease, anorexia, and night sweats.
Tokishakuyakusan
This medicine is used for treating the following symptoms in patients with comparatively weak constitution, slight oversensitivity to cold, and slight anemia who are easily fatigued and sometimes have lower abdominal pain, dull headache, dizziness, shoulder stiffness, buzzing in the ears, and palpitation: menstrual irregularity, abnormal menstruation, painful menstrual periods, climacteric disturbance, general fatigue before/after childbearing or abortion (anemia, fatigue and malaise, dizziness, edema), dizziness, dull headache, shoulder stiffness, low back pain, cold feeling in the lower limbs and waist, chilblain, edema and spots.
Ryokeijutsukanto
This medicine is used for treating the following symptoms in patients with dizziness, light-headed feeling, and pulpitation accompanied by decreased urine volume: nervousness, neurosis, dizziness, heart pounding, shortness of breath, and headache.
Juzentaihoto
This medicine is used for treating patients with declined constitution after recovery from disease, fatigue and malaise, anorexia, perspiration during sleep, cold limbs, and anemia.
Hangekobokuto
This medicine is used for treating the following symptoms in patients who have depressed feelings and a feeling of a foreign body in the throat or esophagus and who sometimes have palpitation, dizziness, and nausea: anxiety neurosis, nervous gastritis, hyperemesis gravidarum, coughing and hoarseness.
Kakkonto
This medicine is used for treating patients with a common cold, coryza, headache, shoulder stiffness, myalgia and pains in the arms and shoulders.
Ninjin’yoeito
This medicine is used for treating patients with declined constitution after recovery from disease, fatigue and malaise, anorexia, perspiration during sleep, cold limbs, and anemia.
Goreisan
This medicine is used for treating the following symptoms in patients with thirst and decreased urine volume who have nausea, vomiting, stomachache, headache or swelling: watery diarrhea, acute gastroenteritis (not used for tenesmus alvi), heat exhaustion, dull headache, and edema.
Rikkunshito
This medicine is used for treating in the following symtoms in patients with a weak stomach, loss of appetite and full stomach pit and those who are easily fatigued, anemic and tend to have cold limbs: gastritis, gastric atony, gastroptosis, maldigestion, anorexia, gastric pain, and vomiting.
Keishibukuryogan
This medicine is used for treating in the following symptoms in patients with a comparatively strong constitution who sometimes have lower abdominal pain, shoulder stiffness, dull headache, dizziness, and feeling of hot flushes with lower limbs being susceptible to cold: menstrual irregularity, abnormal menstruation, painful menstrual periods, climacteric disturbance, automatic imbalance syndrome peculiar to women resembling climacteric disturbance, shoulder stiffness, dizziness, dull headache, contusion, chilblain, and spots.
Reference
Department of Pharmacognosy, National Institute of Health Sciences (NIHS) of Japan and National Institutes of Biomedical Innovation, Health and Nutrition (NIBIOHN). STORK. Available online: http://mpdb.nibiohn.go.jp/stork/index.html (accessed on May 20).
Thank you for your review.

Reviewer 2 Report
This manuscript is a descriptive study analyzing the use of Kampo medicine with a focus on general fatigue at a CAC clinic in Japan. With the post-COVID-19 syndrome likely to become more prevalent and more important in the near future, this topic is clinically important and relevant.
However, I think this manuscript needs further improvement in the following aspects.
- The authors described general fatigue during post COVID-19 syndrome in relation to ME/CFS. However, it is questionable whether ME/CFS should be referred in this manuscript. Because their study data are not related to ME/CFS. Nor do they provide an in-depth review in the context of ME/CFS. Therefore, the authors could justify this by further detailing the association between general fatigue and ME/CFS in patients with long COVID-19 in the Introduction part.
- The authors explained in the Introduction part that there is no firm treatment recommendation for ME/CFS, but there are some clinical guidelines for ME/CFS, including the NICE guideline. Authors will be able to refer it.
- The inclusion criteria of the participants should be more clearly stated. For example, was it considered long COVID-19 syndrome only if the symptoms were present for more than 4 weeks after detection of SARS-CoV-2 negative? Did general fatigue depend on the patient's subjective complaints? Has the severity of fatigue been assessed by NRS or VAS or related assessment tools? Were the results included in the inclusion criteria of this study?
- How was the Kampo medicine used for the patients determined? Can the authors additionally describe the experience, qualifications, etc. of the doctor who prescribed Kampo medicine? On what criteria did the doctor prescribe the Kampo medicines?
- If the doctor who prescribed Kampo medicine used unique diagnostic methods of Kampo medicine, including tongue diagnosis and pulse diagnosis, it would be more beneficial to add those contents to the analysis of this study.
- In Figure 1, general fatigue and accompanying symptoms are listed according to their frequency. However, what the reader is wondering is whether these listed symptoms are really related to the presence or absence of general fatigue. In other words, did the frequencies of the listed symptoms differ in patients without general fatigue?
- The authors described in their Discussion and Conclusion parts as if Hochuekkito could be effective in improving fatigue syndrome after COVID-19. However, as the study cannot confirm the effectiveness of Kampo medicine, as the authors themselves point out, the Hochuekkito effect cannot be confirmed or even estimated.
Author Response
Major revisions’ points:
Reviewer 2:
1. The authors described general fatigue during post COVID-19 syndrome in relation to ME/CFS. However, it is questionable whether ME/CFS should be referred in this manuscript. Because their study data are not related to ME/CFS. Nor do they provide an in-depth review in the context of ME/CFS. Therefore, the authors could justify this by further detailing the association between general fatigue and ME/CFS in patients with long COVID-19 in the Introduction part.
Response 1: Thank you for your kind suggestion. We have added a detailed description of the association between long COVID and ME/CFS and we have stated that viral infections, including COVID-19, are an important entity in ME/CFS and that malaise is a severe symptom in ME/CFS in the Introduction part. In this study, we simply identified the content of Kampo medicine for long COVID in patients with complaints of general fatigue and not the content of prescriptions for ME/CFS patients. However, from previous studies, it is likely that some long COVID patients with general fatigue might transition to ME/CFS. Previous studies have shown that pharmacological therapies such as glucocorticoids, cytokine inhibitors, antibiotics and behavioral therapies such as cognitive-behavioral therapy and graded exercise therapy could be effective in treating ME/CFS in some limited settings. ME/CFS is occasionally a severe and potentially long-lasting condition, and although it is a socially and medically important disease, its cause and treatment are not well established. This study focused on general fatigue in patients with long COVID and the aim of this study was to clarify the prescription of Kampo medicine for this condition. In the introduction part, we have revised the details of long COVID and ME/CFS because we consider that this study will provide a new perspective for future research on ME/CFS as a post-infectious fatigue syndrome.
2. The authors explained in the Introduction part that there is no firm treatment recommendation for ME/CFS, but there are some clinical guidelines for ME/CFS, including the NICE guideline. Authors will be able to refer it.
Response 2: Thank you for pointing this out. As you indicated, we have added currently recognized treatments for ME/CSF (e.g., pharmacological therapies and behavioral therapies) in the manuscript with reference to the NICE guidelines and other previous studies.
3. The inclusion criteria of the participants should be more clearly stated. For example, was it considered long COVID-19 syndrome only if the symptoms were present for more than 4 weeks after detection of SARS-CoV-2 negative? Did general fatigue depend on the patient's subjective complaints? Has the severity of fatigue been assessed by NRS or VAS or related assessment tools? Were the results included in the inclusion criteria of this study?
Response 3: Thank you for your kind suggestion. Patients with long COVID who were still symptomatic 4 weeks after developing COVID-19 were included in this study. This is based on a previous study (reference number 2,3 and 4). We have added this in the methods part. General fatigue as a long COVID symptom depended on patients' subjective complaints, and we did not perform quantitative evaluations such as NRS or VAS. This point is also described in the limitations.
4. How was the Kampo medicine used for the patients determined? Can the authors additionally describe the experience, qualifications, etc. of the doctor who prescribed Kampo medicine? On what criteria did the doctor prescribe the Kampo medicines? If the doctor who prescribed Kampo medicine used unique diagnostic methods of Kampo medicine, including tongue diagnosis and pulse diagnosis, it would be more beneficial to add those contents to the analysis of this study.
Response 4: Kampo medicines were prescribed on the basis of the diagnosis by means of Kampo medicine. Patients’ symptoms are recognized through the basic concepts of qi, blood, and fluid, yin-yang, excess and deficiency, heat and cold, exterior and interior, organ system (so-called heart, liver, kidneys, spleen and lungs) and specific six- staged patterns. In addition, physician could identify any presenting signs specific to the patient’s clinical condition. The diagnosis obtained by this process is called “pattern” (“sho” in Japanese). Not all physicians in the CAC clinic necessarily perform all of the assessments for Kampo diagnosis, including tongue and pulse diagnosis, since these are complex conditions and the diagnosis depends on the physician's experience. However, the physicians who prescribed the Kampo medicine were well-trained general practitioners. They had been practicing general medicine, including Kampo medicine, for 5 to 30 years after graduation. Any physician can consult with doctors specialized in Kampo medicine and all of these doctors often discussed with each other about the long COVID patients at the post COVID condition conferences. We have revised the manuscript and added appropriate information in the limitations part.
5. In Figure 1, general fatigue and accompanying symptoms are listed according to their frequency. However, what the reader is wondering is whether these listed symptoms are really related to the presence or absence of general fatigue. In other words, did the frequencies of the listed symptoms differ in patients without general fatigue?
Response 5: As you point out, the frequency of symptoms in patients without general fatigue is an interesting point. We have changed the description in the Results by designating the frequency of symptoms as Figure 1 with a comparison of frequencies of symptoms in patients with and those without general fatigue. We also revised the discussion based on the results in Figure 1.
6. The authors described in their Discussion and Conclusion parts as if Hochuekkito could be effective in improving fatigue syndrome after COVID-19. However, as the study cannot confirm the effectiveness of Kampo medicine, as the authors themselves point out, the Hochuekkito effect cannot be confirmed or even estimated.
Response 6: Thank you for your suggestion. Since the statement regarding the estimation of the effect of Hochuekito is an overestimation, it has been deleted and revised to a discussion based on the results.
Thank you for your review.
